# Prevalence and Spectrum of HIV-1 Resistance Mutations in the Siberian Federal District

**DOI:** 10.3390/v14102117

**Published:** 2022-09-25

**Authors:** Sergey Shtrek, Lidiya Levakhina, Aleksey Blokh, Oksana Pasechnik, Nataliya Pen’evskaya

**Affiliations:** 1Omsk Research Institute of Natural Focal Infections, 644050 Omsk, Russia; 2Omsk State Medical University, 644050 Omsk, Russia

**Keywords:** HIV, incidence, antiretroviral therapy, mutations, resistance

## Abstract

The Siberian Federal District is among the most affected regions with a high prevalence of HIV-infection and is characterized by high HIV-infection incidence rate and high mortality among the HIV-infected population. HIV drug resistance poses a major threat to public health and is associated with increased mortality, HIV incidence, and cost of epidemic control programs. A total of 1281 samples from HIV-infected patients were sequenced and analyzed with the DEONA and HIVdb Program to assess the prevalence of drug resistance mutations in patients in the Siberian Federal District in 2016–2018. The federal surveillance data obtained from 0.5% of HIV-infected patients during the long-term follow-up care in 2021 were also used. The incidence rate of HIV infection in the Siberian Federal District has declined since 2016: from 135.8 per 100 thousand population to 81.1 per 100 thousand population in 2021. Mutations associated with resistance to NRTI and NNRTI were found in 10.3% of the samples in 2016–2018 and in 28.4% of the samples in 2020. The rising prevalence of drug resistance in HIV-infected patients indicates that it is increasingly important to continuously monitor and improve the approaches to the use of effective treatment regimens.

## 1. Introduction

The Siberian Federal District was characterized by a high incidence rate of HIV infection and high mortality among HIV-infected population for quite a long time [1]. In 2021, the Siberian Federal District accounted for 20.0% of all HIV infections in Russia. The Kemerovo, Irkutsk, Novosibirsk, Omsk, and Krasnoyarsk regions were among the 25 Russian regions with the highest incidence of HIV infection in the population [2].

The development of antiretroviral drugs was among the most important milestones in the struggle against HIV-infection progression and development of AIDS. The main goal of antiretroviral drug therapy is suppressing HIV replication as early as possible [3]. Due to antiretroviral drug therapy (ARVT), it became possible to prolong life expectancy and improve quality of life in HIV-infected patients, as well as limit the spread of infection. In recent years, the HIV treatment coverage has increased significantly. The proportion of HIV-infected patients receiving therapy increased from 1.0% in 2005 to 35.5% in 2017 [4]. HIV drug resistance (HIVDR) can compromise the effectiveness of antiretroviral therapy (ART) in reducing HIV incidence and HIV-associated morbidity and mortality. Minimizing the spread of HIVDR is a critical aspect of the broader global response to antimicrobial resistance [5]. Resistant variants can be transmitted to treatment-naïve individuals, which can lead to rapid virological failure and limit treatment options. Consequently, quantifying the prevalence, emergence, and transmission of drug resistance is critical to effectively treat patients and to shape health policies [6]. Surveillance of HIV resistance to antiretroviral drugs has identified three stages in HIV resistance development across the world. The first stage, before 2000, was associated with the initiation of treatment programs in countries and the increase in prevalence of resistance to nucleoside reverse transcriptase inhibitors. Between 2000 and 2005, there has been an increase in the prevalence of resistance to non-nucleoside reverse transcriptase inhibitors, related to the expansion of treatment mainly using drugs belonging to this class. The past few years have witnessed reduced prevalence of primary (transmissible) resistance due to treatment optimization, on the one hand, and emergence of multiple resistant HIV strains on the other hand [7].

The aim of this study was to analyze the prevalence mutations associated with resistance to protease inhibitors (PI), nucleoside, and non-nucleoside reverse transcriptase inhibitors (NRTI and NNRTI, respectively), among HIV-infected patients in the Siberian Federal District.

## 2. Materials and Methods

This study was based on the results of surveillance for HIV-infection in the Siberian Federal District in 2011–2021. Federal statistical surveillance forms No. 61 “Information about disease caused by human immunodeficiency virus” from the Siberian Federal District were used. Official reporting forms of federal statistical observation were obtained from medical organizations in ten regions of the Siberian Federal District for epidemiological analysis.

The retrospective epidemiological analysis of morbidity, mortality, and prevalence of HIV infection in the Siberian Federal District was conducted according to the conventional algorithms.

A total of 1758 blood samples were collected from HIV-infected patients undergoing antiretroviral therapy in different regions of the Siberian Federal District in 2016–2018 to assess the rate of mutations causing antiviral drug resistance. The study sample involved 19.9% of females (*n* = 416) and 80.1% of males (*n* = 1342).

The study was conducted in the Virology Laboratory of the Siberian Federal District Center for Prevention and Control of AIDS in Omsk.

According to the Russian clinical guidelines, the diagnosis of HIV infection is made by an infectious disease physician based on a comprehensive assessment of epidemiological data, the results of a clinical examination, and laboratory tests. To perform standard screening laboratory examination, antibodies against HIV-1 (HIV-2) and HIV p25/24 antigen are determined simultaneously using diagnostic tests, such as enzyme immunoassay. Confirmatory tests, such as molecular biology techniques (immune blotting or HIV-1 nucleic acid testing), are recommended for confirming the HIV results. If a negative or ambiguous result is obtained by immune blotting after the screening test results were positive, it is recommended that the biological sample is examined using the test kit for detecting p24 antigen or HIV RNA or DNA [8].

RNA isolation, RT-PCR, PCR, and sequencing were performed using the commercial AmpliSens^®^ HIV-Resist-Seq test kit procured from the Central Research Institute of Epidemiology according to the manufacturer’s instructions. Isopropanol was used to remove ddNTP from the fragments. Sequencing was performed on an AB 3500 xl genetic analyzer (Applied Biosystems, Waltham, MA, USA).

A total of 1281 samples (72.9%) of *pol* and *gag* sequences have passed the quality control. Of those, 210 samples (16.4%) belonged to ”naive” patients, while 1071 samples (83.6%) belonged to patients receiving ART.

The electrophoregrams of the sequences were analyzed using the Sequencing Analysis v 3.7 software. The resistance mutations were identified and analyzed using the DEONA and HIVdb Program (Genotypic Resistance Interpretation Algorithm V 7.0, https://hivdb.stanford.edu/hivdb/by-mutations/, accessed on 21 July 2021). The rate of the main resistance mutations in HIV-1 was analyzed using the list from the Stanford Drug Resistance Database [9].

Patients’ compliance was assessed by their attending physicians according to the WHO guidelines by counting the number of pills taken by patients.

In accordance with the WHO guidelines, adherence to therapy means that drugs are taken in strict accordance with the doctor’s prescription, the patient takes the medicine on time, at a dose prescribed by the doctor. The WHO recommends that 95% adherence to therapy is considered to be the threshold value, since it provides the best virologic response to the ongoing therapy. The following levels of adherence to therapy were specified: high (the patient takes ≥95% of prescribed doses of ARVs), medium (85–94%), and low (≤85%).

It is recommended that adherence to therapy is evaluated at each planned visit by counting the number of tablets and filling out a special questionnaire. Counting pills allows one to indirectly determine adherence to therapy according to the number of tablets that were handed out and remained unused (the patient brings all the remaining tablets at the visit). The adherence to therapy is assessed using the formula: (N1 − N2)/N3 × 100%, where N1 is the amount of drug handed out (number of tablets); N2 is the amount of the remaining drug (number of tablets); and N3 is the amount of the drug that the patient was supposed to take for a given period of time (number of tablets). The amounts of each drug individually and the total amount of drugs are quantitatively assessed; the results are documented in the patient’s medical records [8,10].

## 3. Results

A total of 309,925 cases of HIV infection were detected in the Siberian Federal District during all the years of observation. By the end of the observation period in 2021, 268,171 patients (1.5% of the population) were receiving long-term follow-up care at medical institutions. The long-term dynamics of HIV incidence since 2016 showed a decreasing trend incidence rate: from 26,235 cases or 135.8 per 100,000 population in 2016 to 13,839 cases or 81.1 per 100,000 population in 2021 (Appendix A).

In 2021, the percentage of males among the identified HIV-infected patients was 59.3%; the percentage of females was 40.7%, which does not differ from similar indicators in previous years. On average, the male-to-female ratio among the HIV-positive patients in the Siberian Federal District over the past 10 years has been 3:2.

The role of transmission of HIV infection by sexual contact has significantly increased. In 2021, the percentage of heterosexual transmissions in the Siberian Federal District was as high as 73.8%. There was an uneven territorial distribution of the number of HIV cases; the highest prevalence of HIV cases was registered in the Kemerovo (2877.3 cases per 100,000 population), Irkutsk regions (2770.0 cases per 100,000 population); the lowest one was in the Republic of Tyva (118.5 cases per 100,000 population; Appendix A).

The observed territorial distribution of HIV cases is associated with the features of the socioeconomic development of Russian regions, the prevalence of the transmission routes for HIV infection, the duration of the epidemic, the quality of medical care, and the effectiveness of anti-epidemic measures.

During the study period, parameters characterizing the preventive measures taken in the Siberian Federal District have significantly improved. The percentage of population subjected to screening examination for detecting HIV infection has increased. Over the past six years, the percentage of the population screened annually has increased from 22.1% (*n* = 4,280,530) to 28.5% (*n* = 4,812,970) (*p* = 0.00).

The percentage of patients who had interrupted treatment for various reasons decreased by 4.7%, amounting to 5.0% (7205 patients) in 2021 vs. 9.7% in 2016. Among all the patients who had interrupted treatment in 2021, the death rate was 41.6% (vs. 40.2% in 2016).

The effectiveness of antiretroviral therapy, one of the main measures aimed at combating the HIV epidemic, is significantly affected both by the level of screening of the population and by circulation of strains resistant to the antiviral drugs.

An analysis of treatment compliance among the patients included in the study has shown that the largest number of patients (36.3%; *n* = 573) had medium degree of treatment compliance (Table 1).

A total of 29.0% of the samples were found to have mutations associated with resistance to at least one group of ARV (*n* = 372); among them, 15.3 of samples were those with any resistance to NRTIs mutation (*n* = 196), sequences with any NNRTI mutations were found in 13.3% of samples (*n* = 171). Mutations both to NRTIs and NNRTIs were detected in 10.3% of samples (*n* = 132). Sequences with any mutations to PI were found in 1.7% of samples (*n* = 22; Table 2).

The percentage of samples carrying resistance mutations to all three groups of ARVs was low (0.4%).

Over the entire observation period, the main drug resistance mutations affecting the development of HIV-1 resistance to NRTIs and NNRTIs (M184V, K101E, K103N, Y181C, and G190S) occurred with a high rate.

The most common mutations were as follows: M184V (15.9%) leading to high level of resistance to lamivudine and emtricitabine and low level of resistance to didanosine and abacavir; K103N (10.1%) was responsible for a high level of resistance to nevirapine; G190S (5.2%) was responsible for a high level of resistance to nevirapine and efavirenz; K101E (3.0%) was responsible for a medium level of resistance to nevirapine; and A62V (4.2%) was responsible for being an additional mutation that often occurs in combination with other mutations but does not reduce NRTI sensitivity when occurring alone (Table 3).

Mutations causing resistance to protease inhibitors were detected in isolated cases, which was attributed to the presence of a high genetic barrier in this group of drugs.

According to federal surveillance in medical facilities of the Siberian Federal District, 1329 HIV-infected patients underwent resistance mutation sequencing (0.5% of all patients receiving long-term follow-up care) (Table 4).

Among the studied HIV-1 samples, mutations causing resistance to NRTIs were detected in 19.2% of cases (*n* = 255); to NNRTIs, in 14.1% of cases (*n* = 188); to protease inhibitors, in 0.7% of cases; and to integrase inhibitors, in 0.3% of cases. Simultaneous resistance to drugs of two groups, NRTIs and NNRTIs, was detected in 28.4% of patients (*n* = 378).

## 4. Discussion

HIV infection, which has claimed more than 36.3 million lives, continues to be one of the major global public health problems [11].

In 2021, the prevalence of HIV infection in Russia was 754.8 per 100,000 population. HIV infection cases have been registered in all the entities of the Russian Federation. The number of regions with a high prevalence of HIV infection (>0.5% of the total population) is steadily increasing: from 22 in 2014 to 38 in 2020. More than half of the entire Russia’s population (62.3%) and most HIV-positive patients (83.8%) lived in these regions.

In general, the HIV epidemic in Russia was concentrated; however, more than 1% of pregnant women were infected with HIV in 28 Russian regions, which indicated that in these regions it has become generalized according to the WHO classification [2].

Today, expansion of the access to effective prevention, diagnosis, treatment, and care of patients has contributed to the transition of HIV infection from lethal to manageable chronic infection, allowing one to lead a long, active, and relatively prosperous life [3,4,5].

An unprecedented increase in the use of antiretroviral therapy was seen in the past decade. At the end of 2020, 27.5 million of the approximately 37.7 million people living with HIV globally were receiving ART [5].

The expanding access to ART has been accompanied by an increase in HIV drug resistance, which is predicted to be associated with an increase in mortality, HIV incidence, and an increase in the cost of epidemic control programs [12].

HIV drug resistance is caused by changes in the HIV genetic structure that affect the ability of drugs to block viral replication [4,6].

Virtually all modern antiretroviral drugs, including new classes, are at risk of reduced activity against HIV due to the emergence of drug-resistant strains of the virus. HIV drug resistance may compromise the effectiveness of antiretroviral drugs [5,12].

WHO urges countries to monitor resistance and recommends that switching first-line drugs to a more reliable regimen containing dolutegravir after drug resistance to non-nucleoside reverse transcriptase inhibitors (NNRTIs) such as nevirapine and efavirenz reaches a threshold of 10% [5].

The results of a systematic review suggested that patients treated with NNRIT antiretroviral drugs were three times more likely to develop resistance, which reduced treatment effectiveness and highlighted the need to switch to more effective regimens [13].

A number of studies showed that the problem of HIV resistance is also relevant for other Russian regions.

In the Volga Federal District, HIV drug resistance mutations were found in 64.0% (481/752) of the samples of HIV-positive patients in 2016–2018. For each year included in this study, M184V was the dominant substitution (mutation frequency being 26.8%, 22.1%, and 29.3%, respectively). In 2018, the frequency of the K65R mutation increased significantly by 18.0% (0.9% and 5.5% in 2016 and 2017, respectively). The most common NNRTI resistance-associated mutation was the G190S substitution (9.1%, 13.0%, and 20.4%). The frequency of the Y181C mutation increased 3.5-fold compared to the previous years (4.4%, 5.9%, and 16.6%). During the study period (2016–2018), the prevalence of multidrug-resistant HIV strains increased 2.3-fold (51.5%) compared to the period before 2016, thus indicating that there is an unfavorable prognosis for the development of the epidemic in the Volga Federal District [14].

In the Ural Federal District, at least one drug resistance mutation was found in 140 out of 223 samples (62.8%). NNRTI and NRTI resistance mutations were the most common ones, occurring in 53.4% (*n* = 119) and 47.1% (*n* = 105) of all the samples, respectively. MDR to PV was registered much less frequently: in 11.7% (*n*= 26) cases. M184I amino acid substitutions conferring resistance to both NRTIs and NNRTIs were found in nine samples (4.0%). Combinations of mutations simultaneously conferring resistance to NRTIs and NNRTIs were detected in 36.3% of the samples (*n* = 81). MDR to all classes of ARVs was detected in 4.9% of the samples (*n* = 11). The most common one is the M184V amino acid substitution, which confers HIV resistance to NRTIs; this mutation was more common than any other MDR mutations and was detected in 88 samples (39.5%). Among mutations responsible for MDR to NNRTIs, the G190S amino acid substitution was the most common one (34.7%, *n* = 55). The most common HIV-1 MDR to PI was the minor mutation L33F detected in 13 samples (5.8%) and M46I observed in 9 samples (4.0%). Although detected least frequently, MDR to PI was characterized by a wide variety of amino acid substitutions: 23 variants (33.3% of all MDR variants) [15].

In the Far Eastern Federal District, drug resistance mutations were detected in 48% (24/50) of HIV-positive patients receiving antiretroviral therapy. A HIV strain simultaneously resistant to two classes of drugs (NRTIs and NNRTIs) was detected in 28% of patients (*n* = 14). The highest resistance level was found for NRTI drugs (emtricitabine and lamivudine) and NNRTI drugs (nevirapine and efavirenz). The most common amino acid substitutions were M184V (22.0%), K101E (14.0%), K103N (12.0%), and G190S (12.0%) [16].

Our study conducted in the Siberian Federal District, one of the most affected regions with a high prevalence of HIV infection, has revealed a moderate trend towards decreasing incidence of HIV infection among the population, which is associated both with the effectiveness of preventive programs and with the increasing coverage of patients with antiretroviral therapy.

Meanwhile, the problem of HIV drug resistance to antiretroviral drugs still remains. Most of the examined patients had an average level of treatment compliance, which is a factor contributing to the development of resistance. Most of the sequences obtained from patients receiving ART carried mutations affecting the sensitivity to NRTIs and NNRTIs. A62V, K103N, M184V, G190S, and K101E were the most common mutations, which are associated with resistance to drugs of NRTI groups (lamivudine and emtricitabine) and NNRTIs (nevirapine and efavirenz).

The growing prevalence of drug resistance in HIV-infected patients demonstrates that it is important to continuously monitor and improve the approaches to using effective treatment regimens.

Although the slight reduction of the number of newly detected HIV infection cases and the number of deaths can be associated with measures related to covering a larger number of patients with treatment, the overall epidemiological situation indicates that these measures are insufficient for preventing the spread of HIV in the Siberian Federal District.

Our findings showed that it is necessary to improve the quality of epidemiological surveillance of HIV drug resistance, optimize the approaches to ensuring patients’ adherence to therapy, and improve the efficiency of follow-up monitoring and treatment for HIV-positive patients.

## Figures and Tables

**Table 1 viruses-14-02117-t001:** Treatment compliance among patients included in the study (%).

Compliance to Antiretroviral Therapy			
High	Medium	Low	Unknown
255 (14.5%)	656 (37.3%)	296 (16.8%)	551 (31.3%)

**Table 2 viruses-14-02117-t002:** Samples of HIV-1 with resistance mutations in the Siberian Federal District in 2016–2018, absolute number, and %.

Resistance to	N, abs.	Percentage, %
At least one ARV	372	29.0
Any PI	22	1.7
Any NRTI	196	15.3
Any NNRTI	171	13.3
Any NRTI and NNRTI	132	10.3
Any NRTI and NNRTI and PI	5	0.4

**Table 3 viruses-14-02117-t003:** The rate of the main drug resistance mutations in 2016–2018 (%).

NRTI Mutations, %								
A62V	D67N	K70R	M184V	M41L	L74V	T215F	K219Q	V75M
54 4.2	11 0.8	13 1.0	204 15.9	26 2.0	29 2.2	6 0.5	11 0.8	3 0.2
NNRTI mutations, %								
K101E	K103N	G190S	G190C	A98G	H221Y	Y181C	Y188L	P225H
38 3.0	130 10.1	67 5.2	4 0.3	3 0.2	8 0.62	30 2.3	1 0.07	14 1.1
PI mutations, %								
L33F	K43T	M46I	I54M	I50V	N88D	V82A	I184V	
1 0.07	1 0.07	11 0.8	3 0.23	1 0.07	1 0.07	1 0.07	1 0.07	

**Table 4 viruses-14-02117-t004:** HIV-1 resistance mutations in the Siberian Federal District in 2021, absolute number and %.

Resistance to	N, abs.	Percentage, %
NRTI	255	19.2
NNRIT	188	14.1
PI	9	0.7
INSTI	4	0.3
NRTI and NNRTI	378	28.4
NRTI and PI	22	1.7
NNRTI and PI	1	0.1
NRTI and INSTI	0	0.0
NNRTI and INSTI	2	0.2
NRTI, NNRTI and PI	17	1.3
NRTI and NNRTI and PI and INSTI	0	0.0
Other	4	0.3
Total	1329	

## Data Availability

Not applicable.

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
