# Peer review of "Prevalence and Spectrum of HIV-1 Resistance Mutations in the Siberian Federal District"

_viruses, 2022, doi:10.3390/v14102117_

Round 1
Reviewer 1 Report
Drug resistance of mutations in HIV-1 is the most severe challenge to the long-term efficacy of HIV-1 inhibitors in highly active antiretroviral therapy. In this work, the authors investigate the incidence rate of HIV infection in the Siberian Federal District in the last five years. Although the incidence rate has declined since 2016, the drug resistance to mutation has increased. This is an impressive brief study of HIV-1 resistance mutations in a given region. However, I have the following comments and questions that need to be addressed before the manuscript is published.
1. The background of this work is not sufficient, especially for the drug resistance of mutations.
2. In the “Materials and Methods” section, some key informations are missing, such as in the last paragraph of this section, the detail about the WHO guidelines.
3. The authors show the uneven territorial distribution in figure 2, which want to indicate what? More deep analyses are expected to explain this uneven distribution.
4. It’s curious for the discussion section, per paragraph has almost one sentence.
5. The figures are expected to improve the quality.
Author Response
"Please see the attachment."

Reviewer 2 Report
In the manuscript “Prevalence and spectrum of HIV-1 resistance mutations in the Siberian
federal district”, the authors reported an epidemiology study of HIV-1 drug resistance mutations detected in the Siberian federal district of Russia. Here are my comments and suggestions.
1. The authors reported the overall HIV-1 epidemic statistics over the past 10 years in the Siberian federal district, which provides great information for the users to understand the background of HIV-1 epidemic in the area. However, the authors need to describe how these numbers were obtained in the methods section and briefly describe the HIV-1 diagnosis assay they used in the area.
2. Line 47, the authors stated that the samples were collected from those on antiviral therapy. Did they fail the antiviral therapy and went for a resistance testing? What were the antiviral regimens they used? Transmitted drug resistance mutations (from treatment naïve patients) look very different from those who fail the therapy. The authors also used the number of DRMs from the data in 2021 (Table 4). It is unclear how these DRMs data were collected and what population was targeted (treatment naïve or treatment experienced).
3. Line 50-51, these data are better to be described in the Results section.
4. Any viral load information of the samples that had DRM testing?
5. Line 82, 118.0 cases per 100,000 population in the text. But in the Fig. 2, the number is 118.5. Please double check.
6. Line 89 to 91, the authors need to add p value in the comparison if they claim the differences are significant.
7. Line 93 to 94. It is unclear the meaning of death rate here. Because the longer the follow-up is, the more deaths will be observed. The authors may want to use mortality (person-year) here.
8. Table 1. What is the definition of High Medium and low Compliance to antiviral therapy? Also I don’t think adding a column of Study period is helpful here because there is only one study period.
9. Table 2. Any IP should be Any PI
10. Table 3. As in Table 1, I don’t think adding a column of Study period is helpful here because there is only one study period.
11. The authors may want to discuss in depths how their findings would impact clinical management and prevention of HIV-1 in the area.
Round 2
Reviewer 1 Report
Thanks for the revision. No more comments.
Reviewer 2 Report
The authors have addressed my comments.